# Origin Firing Regulations to Control Genome Replication Timing

**DOI:** 10.3390/genes10030199

**Published:** 2019-03-06

**Authors:** Dominik Boos, Pedro Ferreira

**Affiliations:** Vertebrate DNA Replication Lab, Centre for Medical Biotechnology, University of Duisburg-Essen, 45141 Essen, Germany; pedro.ferreira@uni-due.de

**Keywords:** eukaryotic origin firing, origin firing regulation, origin firing timing, replication timing

## Abstract

Complete genome duplication is essential for genetic homeostasis over successive cell generations. Higher eukaryotes possess a complex genome replication program that involves replicating the genome in units of individual chromatin domains with a reproducible order or timing. Two types of replication origin firing regulations ensure complete and well-timed domain-wise genome replication: (1) the timing of origin firing within a domain must be determined and (2) enough origins must fire with appropriate positioning in a short time window to avoid inter-origin gaps too large to be fully copied. Fundamental principles of eukaryotic origin firing are known. We here discuss advances in understanding the regulation of origin firing to control firing time. Work with yeasts suggests that eukaryotes utilise distinct molecular pathways to determine firing time of distinct sets of origins, depending on the specific requirements of the genomic regions to be replicated. Although the exact nature of the timing control processes varies between eukaryotes, conserved aspects exist: (1) the first step of origin firing, pre-initiation complex (pre-IC formation), is the regulated step, (2) many regulation pathways control the firing kinase Dbf4-dependent kinase, (3) Rif1 is a conserved mediator of late origin firing and (4) competition between origins for limiting firing factors contributes to firing timing. Characterization of the molecular timing control pathways will enable us to manipulate them to address the biological role of replication timing, for example, in cell differentiation and genome instability.

## 1. A Simplified Model of Genome Replication in Higher Eukaryotes

### 1.1. Completeness of Genome Replication Requires Small Distances between Origins and Reliable Origin Firing

Genome replication must be complete and accurate to generate two faithful copies of the DNA in order to inherit a full set of genes to both daughter cells. Completeness of genome duplication is a particularly complex issue in eukaryotes that require multiple replication origins to duplicate their large genomes within a typical replication period that can last from several minutes in yeasts to some hours in vertebrates. With multiple origins comes the risk of gaps between two replication initiation sites that are too large to be replicated by two replication forks within the duration of an S phase. In order to keep the number of large inter-origin gaps minimal to guarantee completeness of genome duplication two requirements must be met (Figure 1A): (1) origins must be placed regularly, not too far from each other in the genome and (2) origins must fire reliably as any unfired origin increases gap size.

In most eukaryotes including metazoa the large-gap problem becomes aggravated by the fact that origin positions are not strictly determined by DNA sequence. This generates an element of randomness in the location of origins and, thus, a certain size distribution of small and large inter-origin distances [1]. The larger the genome, the higher the number of origins must be at a given replication fork speed and mammalian genomes have several ten thousand origins. Because larger genomes have more origins the total number of very large inter-origin distances that can potentially not be completely replicated rises with genome size.

One important mechanism utilized by eukaryotes to reduce large inter-origin distances that is not discussed in detail in this article is the use of dormant origins [2]. Eukaryotic cells generate many more potential origins by origin licensing (discussed below) than actually fire in normal S phases [3,4,5,6,7,8]. These are dormant origins that fire only if they have enough time, as shown in yeast and vertebrates [9,10,11]. Unfired dormant origins will normally get inactivated if their DNA is replicated by an incoming fork generated by a neighbouring origin. If this does not occur, for example when no origin in the vicinity fires or when DNA damage stalls incoming forks, dormant origin firing helps complete duplication of the DNA in their genome region.

In addition to dormant origins that help reduce size of many inter-origin gaps eukaryotes have also evolved a complex genome replication program to ensure complete genome replication, which is discussed in detail in this review.

### 1.2. Domain-Wise Replication May Help Complete Genome Duplication in Limited Origin Firing Conditions

Some origin firing factors are present in limiting amounts in somatic eukaryotic cells [12,13,14,15]. Hence, not all origins can fire at the same time at the beginning of S phase but must fire distributed over the duration of S phase. If the relatively rare firing events were scattered throughout the genome extremely large inter-origin distances would ensue. Consequently, the likelihood of persistently stalled forks that cannot be rescued by an incoming opposing fork would be high and so would be the rate of fork breakdown and genomic re-arrangements [9,10,11,16]. Presumably to avoid this, higher eukaryotic cells concentrate origin firing locally, which leads to genome replication in the form of separate genomic domains, called replication domains/factories [17]. This domain-wise replication may represent a way to make replication locally efficient within replication domains, despite low overall origin firing.

### 1.3. Timing of Domain Replication

The timing with which replication domains replicate is an important aspect of eukaryotic genome duplication. Replication timing changes with cell differentiation, as shown by genome-wide replication timing analyses and these timing changes are conserved between different human and mouse cell types, indicating that replication timing is evolutionarily important [18,19]. Moreover, late replication is associated with elevated levels of genetic alterations [20,21] and chromosome fragility [22].

Replication domains juxtaposed in the genome appear to mostly replicate successively, meaning once a domain has replicated its neighbour starts (dominos model) (Figure 1B) [23,24,25]. This may minimise the frequency with which domains that replicate in isolation occur, that is domains replicating at vastly different times from their neighbours. Such isolated domains inevitably generate two forks, the two outmost forks, that will find opposing forks for termination only with a significant time delay. These forks will therefore have a high probability of permanent stalling. Exceptions to the dominos-like propagation of replication are timing transition regions (TTRs), the regions between two neighbouring domains that replicate with a significant delay. Indeed, TTRs were found to be replicated unidirectionally, consistent with an outmost fork travelling out of its domain (slightly more complex scenarios were suggested) [18,26]. Whether TTRs pose a particular problem for accurate genome replication is a matter of debate.

From this dominos model of higher eukaryotic genome replication follows that the timing with which replication domains replicate is dictated by the replication time of the first replicating domain combined with the rate of spreading of replication through the neighbouring domains (Figure 1B). At TTRs, blocks to replication spreading must exist. The replication timing of the later replicating domain at TTRs must be determined anew (Figure 1C).

This model of genome replication presented is highly simplified and leaves open many questions. What determines the timing of the first domain in a row (the first domino)? How is spreading of replication achieved and how is it blocked at TTRs? Which chromatin processes and parameters influence replication timing?

Some fundamental aspects of replication timing in eukaryotes have been characterised. It has long been known that transcriptionally active open euchromatin replicates in early S phase, whereas the structurally compact heterochromatin replicates late (with some exceptions). Microscopically, early replication becomes visible in mammalian cells as hundreds of foci of active replication scattered throughout the nucleus after pulse-labelling with detectable nucleotide analogues or when expressing labelled PCNA (proliferating cell nuclear antigen) [17,27,28,29,30]. Mid and late S phase replication sites are localised close to heterochromatin, at the nuclear periphery and around nucleoli. Very late S phase replication is visible as few large bright foci. The microscopic replication domains were estimated to contain about 1 Mbp of DNA based on the number of foci per cell and genome size [29]. At least six replicons (12 forks) are required to replicate such domains, as calculated from the average replication fork speed of 1.2 kbp/min and a life time of a typical factory of about 45 min [29]. Genome-wide replication timing analyses confirmed similar mean replication domain sizes of 1.4–3.6 Mbp, albeit with a high variability [18]. Domains that changed replication timing during differentiation had a lower average size of 0.4–0.8 Mbp. Active replication and transcription occur largely separately from each other, as co-labelled active replication and transcription foci showed little overlap in S phase nuclei [31,32,33]. This separation could help minimise replication-transcription interference to avoid genome instability [34,35,36,37].

Interestingly, some genomic regions are subject to specific timing regulation. The heterochromatin at pericentromeres replicates very early in budding and fission yeasts, as does the heterochromatic mating type locus in fission yeast, whereas other heterochromatic regions replicate late. There are specific molecular mechanisms to determine these early and late replication times [38,39], providing first examples that eukaryotes use active regulation to facilitate the appropriate replication timing. We will discuss these examples in detail below.

## 2. Complete and Well-Timed Domain-Wise Genome Replication Requires Coherent Origin Firing within Domains and Proper Control of Origin Firing Time

Two types of control of replication origin firing are necessary in order to avoid large inter-origin gaps and ensure appropriately timed replication in the described domain-wise replication mode: (1) The timing with which the origins within individual domains fire needs to be controlled and (2) once a domain is set to replicate its origins must fire coherently—efficiently and in a narrow time window—to be fully replicated in the typical 45–60 min [17,30,40] a domain takes to replicate. To illustrate that coherent firing matters: A typical 1 Mbp replication domain can be copied by about 16–22 forks from 8–11 origins at a fork speed of 1 kb/min in 45–60 min if all origins fire reliably at the same time. If 50% of the origins fired with a delay of 30 min of a 60 min domain replication period 240 kbp will be left unreplicated if not compensated by firing of at least four more origins after 30 min.

## 3. Molecular Mechanisms of Origin Firing

All temporal and spatial regulations of replication origin firing must ultimately control the activity of essential origin firing factors. The molecular mechanisms of eukaryotic origin firing are relatively well-known in budding yeast, where genetic screening, cell biology and biochemistry identified the essential origin firing factors and characterised fundamental activities and regulations. Recent biochemical re-constitution experiments then defined the set of core firing factors sufficient for in vitro origin firing, the establishment of bi-directional DNA replication with leading and lagging strand synthesis out of pre-replicative complex (pre-RCs) [41]: the Mcm2-7 helicase, the cell cycle kinases CDK and DDK, Sld2, Sld3, Sld7, Dpb11, Cdc45, the GINS complex, DNA polymerases epsilon and α/primase and Mcm10.

Eukaryotic replication initiation has two steps, origin licensing and origin firing. Licensing is the loading of the replicative helicase comprising the six Mcm2-7 subunits onto origin DNA. Licensing occurs exclusively under low-CDK conditions during mitotic exit and in the G1 phase of the cell cycle [42]. The loaded helicase forms a head-to-head dimer of two Mcm2-7 hexamers that encircle double-stranded DNA [43,44] (Figure 2A). The loaded Mcm2-7 double hexamer is called pre-RC. The core licensing machinery sufficient for pre-RC formation on naked origin DNA in vitro constitutes ORC (origin recognition complex), Mcm2-7, Cdt1 and Cdc6 [43,44]. Other factors influence licensing of chromatin in living cells, such as the chromatin factors Hbo1 [45,46], SNF2 [47], GRWD1 [48] and PR-Set7 [49,50,51,52,53] or Mcm-BP [54,55]. In the pre-RC the helicase is not active, as no DNA unwinding seems to occur. Origin firing activates the helicase and converts pre-RCs into two active bi-directional replication forks or replisomes [56,57,58,59]. Because origin firing requires high CDK and DDK activities it can only occur from G1-S transition when these kinases become active. In vitro, the first step of origin firing is phosphorylation of pre-RCs by DDK, which leads to the binding of the Sld3-Sld7 protein complex [60,61,62,63,64] (Figure 2B,C). In vivo, the situation might be slightly more complicated as Sld3 and Cdc45 recruitment to origins depend on each other [65,66]. CDK phosphorylates Sld3 and Sld2 that constitute the minimal set of CDK substrates sufficient for replication [67,68,69,70] (Figure 2D). Phospho-Sld3-Sld7 and phospho-Sld2 bind to separate phosphorylation and sequence-specific protein binding domains in the Dpb11 protein, BRCT repeat domains (BRCA1 C-terminal repeats) [69,70] (Figure 2D). As a result, the so-called SDS complex (Sld3/7-Dpb11-Sld2) may form. Although phospho-Sld2 and phospho-Sld3 interactions with Dpb11 events are essential for origin firing [69,70] it is possible that, in vivo, they do not bind simultaneously or bind transiently because a stable SDS complex in cells has not been reported. Sld2 associates with and recruits GINS and DNA polymerase epsilon to pre-ICs [71], whilst Sld3 recruits Cdc45 [63]. We here call the SDS complex, Cdc45, GINS and polymerase epsilon bound to pre-RCs the pre-initiation complex (pre-IC) [66,72], although the exact structure of the pre-IC remains unclear. Pre-IC can be stably assembled in vitro [41,64] and can be detected by ChIP in vivo [66] (Figure 2E). Cdc45 and GINS assemble stably [73] with the Mcm2-7 helicase to form the CMG complex (Cdc45-Mcm2-7-GINS) [56,74], the active replicative helicase [57,75] (Figure 2F).

During CMG formation the SDS complex proteins dissociate from origins [73]. In contrast to the other origin firing factors the SDS proteins are thought to have no role in active replisomes but constitute a regulation platform of origin firing. Once they have mediated CDK and DDK-dependent origin firing they are not needed any more and can help fire the next origin.

After CMG formation active replisomes assemble in an Mcm10 dependent manner [41,76,77,78], which involves additional replication proteins and complex re-arrangements of the CMG helicase and the origin DNA [58,79] (Figure 2G). Bi-directional DNA synthesis can commence.

Although little mechanistic detail of replication initiation in metazoa is known, many of the basic principles of origin firing appear to be conserved, including those of the SDS firing regulation platform. Like in yeast, metazoan Sld3, Treslin/TICRR^Sld3^ [80,81,82], mediates CDK-dependency of origin firing by phosphorylation-mediated interaction with the Dpb11 orthologue TopBP1 [83,84]. Vertebrate Sld2, RecQL4, is an essential firing factor [85,86] but does not seem to depend on CDK. At first, an orthologue of the Sld3^Treslin^ interactor Sld7 [87] could not be identified in metazoa. Instead, an essential origin firing factor, the MTBP protein, for which no orthologue in yeast could be found and that binds to Treslin/TICRR^Sld3^ was described [88]. Only later remote homology of MTBP with Sld7 in two protein regions could be detected [89]. MTBP^Sld7^ completed the list of core origin firing factors in higher eukaryotes that had been defined in yeast. A stable protein complex containing Treslin/TICRR^Sld3^, MTBP^Sld7^ and TopBP1^Dpb11^ could be detected in cell lysates [88]. We will call it TMT complex (Treslin-MTBP-TopBP1) from here.

## 4. Pre-IC Formation is a Main Regulation Step of Origin Firing

Many cellular molecular pathways are regulated at an early step to guarantee an economical use of resources. Naturally, the regulated activity in a molecular pathway must be rate limiting. The pre-IC factors DDK, Sld2/RecQL4, Sld3/Treslin, Dpb11/TopBP1 and Cdc45, have been shown to be limiting in yeast and/or vertebrate model systems [12,13,14,15]. Consistent with pre-IC formation (Figure 2B–E) being a regulation step of origin firing, the described cell cycle regulations by CDK and DDK occur at the pre-IC formation stage in yeasts and metazoa [60,62,63,64,69,70,83,84,90]. Regulation at the level of pre-IC formation is not a cell cycle-specific feature but rather a principle of origin firing control, as exemplified by the fact that firing regulation upon DNA damage also invokes pre-IC factors. Under DNA damage conditions, origin firing becomes inhibited by the S phase checkpoint to prevent genetic alterations due to copying damaged DNA templates. In yeast, the checkpoint inhibits CDK and DDK action [91,92,93]. The Rad53 checkpoint kinase phosphorylates Sld3^Treslin^, suppressing the CDK-dependent binding of Sld3^Treslin^ to Dpb11^TopBP1^. Rad53 also phosphorylates the Dbf4 subunit of DDK, which inhibits this kinase. As discussed in detail below, the origin firing timing mechanisms uncovered to date also involve the regulation of pre-IC factors.

The metazoan orthologues of the SDS complex, TMT complex proteins Treslin/TICRR^Sld3^, MTBP^Sld7^ and TopBP1^Dpb11^, have metazoa-specific protein domains in addition to the core domains conserved with yeast [88,89]. These metazoa-specific domains may well constitute regulatory units that mediate proper control of origin firing in the complex metazoan cells, in which appropriately controlled origin firing is probably particularly important. Consistently, the metazoa-specific C-terminus of Treslin/TICRR^Sld3^ has non-essential roles in replication [94,95], as discussed below and the metazoa-specific MTBP^Sld7^ middle region is required for replication [89]. This MTBP^Sld7^ middle domain role partly depends on binding the Cdk8/19-cyclin C kinase, although the exact role of this interaction is unclear.

The SDS/TMT regulation hub alongside CDK and DDK kinases will certainly prove to have a key role in controlling both timing and coherence of origin firing to achieve complete and well-timed domain-wise genome replication.

## 5. Origin Firing Control to Determine Replication Timing

How is the timing of origin firing in individual replication domains determined? Two principle possibilities appear plausible: (1) chromatin organisation in structural units determines if and when a domain fires its origins. For this, origin firing factors, for example TMT, CDK or DDK, need to be granted access to or allowed to be active within the confines of the domain at the right time (Figure 3A). (2) Firing of clusters of origins is regulated at the origin level rather than being a consequence of a structural chromatin unit. For this, the origins in a cluster need to be regulatorily coupled. Coupling could involve that the origins in a cluster are subject to the same control pathway, be it inhibitory so that origins are kept inactive until a signal relieves the inhibition or activating so that the firing of neighbouring origins is simultaneously stimulated (Figure 3B).

Intuitively, origin coupling could involve that neighbouring origins influence each other. The unfired origins in a cluster could inhibit each other until at least one origin fires, relieving the inhibitory signal. Alternatively, firing of the first origin in a cluster could create an activating signal and stimulate firing of its neighbours.

Alternatively, timing could be controlled by an intrinsic property of origins rather than by active regulation by external factors. For example, origins of early domains could respond more sensitively than late origins to limiting origin firing factors, for example due to higher accessibility or higher affinity and therefore fire earlier. It was proposed that the different firing probabilities between origin clusters resulting from such specific sensitivities to firing factors could largely explain replication timing programs like those seen in eukaryotes [96].

Models 1 and 2 (Figure 3) are not mutually exclusive. For example, origin coupling could involve elements of chromatin structure and modification. Conversely, origins could modify their chromatin environment depending on their firing activity.

### 5.1. Structural Chromatin Units May Underlie Replication Timing

That stable chromatin units may underlie replication timing was proposed based on the finding that replication in metazoan cells occurs in form of individual physical chromatin units, today called replication domains. Early studies analysing isolated DNA fibres from mammalian cells on which replication had been labelled using radioactive nucleotides showed that origins do not fire scattered over the DNA but that juxtaposed origins often fire nearly simultaneously in clusters (reviewed in Reference [17]). Microscopic work mentioned above using fluorescence labelling of active replication sites established that replication occurs in discrete replication foci in mammalian nuclei. The foci represent a single or a small number of origin clusters. Replication domains were proposed to form stable physical units as labelled domains were stable over many successive cell divisions [29,40,97,98].

A specific replication timing program was proposed based on the observation that the same replication foci patterns (specific for early, mid and late S phase) re-occurred in every S phase [17,27,28,30]. Elegant nuclei transfer experiments then drew a link between chromatin domain formation and replication timing [28]. This work showed that a time point in early G1 phase exists at which replication timing is determined, the timing decision point (TDP). For these experiments, nuclei isolated from synchronised mammalian cells were replicated in *Xenopus* egg extracts. In nuclei isolated from cells in mitosis or G1 before the TDP (up to 1 h after anaphase onset), the different genome regions did not replicate in a defined order but in a random fashion typical for *Xenopus* embryonic extracts. In contrast, chromatin isolated more than 2 h after mitosis replicated in the same order as in the cells of origin. They had passed the TDP. The TDP coincided with the time of re-establishment of an interphase-like chromatin architecture out of the mitotic chromatin. The authors therefore suggested that the establishment of interphase chromatin domains in G1 may specify replication timing in the subsequent S phase. Later genome-wide proximity studies of genome regions in cells by HiC showed a correlation of genome structure with replication timing [19,99]. It turned out that replication domains largely overlap with stable chromatin folding units, topologically associated domains (TADs) [100]. Re-formation of these TADs after mitosis coincided with the TDP [101].

However, direct poof that the structuring of chromatin into folding units underlies the determination of replication timing has not been provided. It has also not been proven that the formation of the microscopically visible replication foci that reflect structural chromatin domains is required to determine replication timing. In fact, genome structure and replication timing do not always correlate: G2 cells retain the overall TAD organisation but replication timing is random when G2 nuclei are forced to replicate either in *Xenopus* egg extracts or by inducing a second replication round in G2 cells [101,102]. Conversely, G0 cells whose chromatin undergoes great changes in organisation retain replication timing. Taken together, it seems that even if the formation of stable chromatin folding units is required to determine replication timing it is not sufficient. One or more activities that are absent in G2 chromatin are required at the TDP for establishment of replication timing.

### 5.2. How Could the Folding of Chromatin into Physical Units Determine Origin Firing Time?

A chromatin domain could form a confined space that excludes or concentrates origin firing factors, thereby controlling firing timing. However, there is little direct evidence to verify this idea.

A well-established concept is that chromatin structure determines the accessibility of its DNA to DNA binding proteins. Controlled accessibility of DNA for firing factors within a chromatin domain could regulate firing timing. Correlations between high DNA accessibility and early replication activity have been drawn. Genome-wide HiC analysis in cultured cells revealed a good correlation between the nuclear compartment containing open, transcriptionally active chromatin and early S phase replication, whereas the compartment containing closed heterochromatin replicates late [19]. Moreover, opening chromatin structure by deletion of histone deacetylases from yeast cells, by recruiting acetylases to chromatin in human cells or by induction of transcription in *Drosophila* can lead to earlier origin firing [103,104,105,106,107]. Recently, it was suggested that more open chromatin induced by preventing methylation of lysine 4 of histone 4 in cultured mammalian cells increases origin firing [108]. Here, origin licensing in addition to origin firing was elevated upon induced chromatin opening, indicating that the amount of licensing could affect whether and how efficiently an origin fires. Perhaps increased pre-RC levels locally increase the concentration of firing factors.

Another model for how chromatin domain formation determines firing timing is that domains could constitute structural units to control DNA position in the nucleus. Re-positioning of domains could move DNA between nuclear regions with high or low concentrations of firing factors. It was suggested that localisation of late replicating telomeric DNA close to the nuclear periphery may withdraw it from regions with high firing factor concentrations in the nuclear interior [109]. However, artificial peripheral localisation is not always sufficient to mediate late replication of a genome region that is normally located in the nuclear interior [110]. Folding of DNA into chromatin domains could also control firing timing by bringing origins in close proximity to each other, as suggested for how forkhead transcription factors mediate early origin firing [111] (discussed in detail below).

### 5.3. In Yeast, Several Origin Firing Control Pathways in Combination with Competition between Origins for Firing Factors Determine Replication Time

It has emerged in recent years that yeast cells utilize distinct molecular pathways to control origin firing timing of different genome regions. There is even more than one pathway to control the firing of separate groups of early and late origins, respectively. For example, both pericentromeric heterochromatin and a large proportion of the transcribed euchromatin replicate early. Yet pericentromeric origins require the kinetochore protein Ctf19 for early firing whereas many euchromatic origins rely on forkhead transcription factors [38,111]. Different late origins require Rif1 and/or Taz1 and/or shelterin for late firing, depending on whether they are in chromosome arms, subtelomeres or telomeres [112,113,114]. These pathways are discussed in detail below.

In addition to such active firing control pathways, competition between origins for limiting firing factors is important to determine firing time. In mutant yeast and mammalian cells, in which firing of early origins is delayed, the firing of later origins is advanced and vice versa [111,115]. Apparently, early origin firing delays late origins in wild-type conditions. In the delay mutants, the early origins do not effectively compete with late origins for firing factors, giving late origins better access to these factors.

Together, these insights suggest that eukaryotes employ several distinct regulation pathways specific for the different genome regions to advance the firing of some origins and delay others. Combined with competition for limiting firing factors these pathways establish the replication timing program (Figure 4). What necessitates several specific molecular mechanisms to determine firing time? To date, we can only speculate that the distinct chromatin structures and the different ongoing chromatin processes in the different genome regions, for example transcription activity, require distinct regulatory pathways.

### 5.4. Forkhead Transcription Factors Advance Origin Firing Timing in Budding Yeast

The forkhead transcription factors Fkh1 and Fkh2 are required in budding yeast for early replication of about 30% of origins [111] (Figure 4). This was revealed by analysis of genome-wide replication dynamics using purified nascent DNA in yeast mutants lacking Fkh1/2. The control of replication timing by forkhead proteins appears to be independent of their classic role in transcription. To mediate early firing the binding of forkhead to DNA in the vicinity of the origin was required. This forkhead-DNA interaction may facilitate the formation of conglomerates of early firing origins, as suggested by DNA proximity analysis using the 4C technique to compare wild-type and forkhead mutant cells [111]. An interaction between Fkh1/2 bound to origins and the licensing factor ORC bound to neighbouring origins may bring juxtaposed origins into proximity. How origin proximity advances firing time is unknown. A conceivable model is that in conglomerates origins cooperatively concentrate limiting firing factors.

In addition to these structural effects of forkhead factors on chromatin Fkh1/2 also directly recruit DDK kinase to early origins to control their firing [116]. Early firing required a direct interaction between Fkh1/2 and the Dbf4 subunit of the DDK kinase. A binding-deficient Dbf4 mutant delayed origin firing. Origin localisation of DDK was the only function of the interaction domain of Dbf4 required for early replication since the firing delay of this mutant was rescued by fusing it to forkhead. Whether and how the structural and DDK recruitment roles of Fkh1/2 cooperate to mediate early firing remains to be investigated.

### 5.5. Early Firing of Pericentromeric Heterochromatin Origins Depends on the Recruitment of DDK by the Ctf19 Kinetochore Protein or Swi6 in Budding and Fission Yeast, Respectively

Despite the typically late replication of heterochromatin, the heterochromatin at pericentromeres replicates in early S phase in yeasts (Figure 4). In fission yeast, pericentromeric heterochromatin as well as the heterochromatic *matK* locus but not the subtelomeric heterochromatin, show early origin firing. Pericentromeres and *matK* recruit DDK dependently on the heterochromatin protein Swi6, the fission yeast orthologue of metazoan heterochromatin protein 1 (HP1) [39]. DDK recruitment is mediated by an interaction between Swi6 and Dfp1, the fission yeast Dbf4 orthologue. Although subtelomeres have also Swi6 bound, their origins fire late. This late firing appears to be due to limiting activity of DDK because artificial recruitment of DDK to subtelomeres advanced replication timing. Whether low concentration of DDK at subtelomeres (despite Swi6) or high subtelomeric activity of the DDK counteracting activity of Rif1-PP1 (discussed below) are responsible for late firing of subtelomeric origins is unclear.

In budding yeast, early replication of centromeres relies on DDK recruitment to pericentromeric origins dependently on the Ctf19 kinetochore protein [38]. Thus, fission and budding yeasts utilise separate molecular mechanisms to fire pericentromeric origins early, both culminating in the recruitment of DDK.

Ctf19-dependent DDK recruitment to pericentromere origins occurs in telophase and G1 phase, indicating that DDK recruitment precedes the firing of the respective origins in the subsequent S phase [38]. This is reminiscent of the above-mentioned TDP in G1 phase described in higher eukaryotes [28].

The presence of DDK at origins in G1 potentially poses a problem. If the G1-recruited DDK was active, the cell would run the risk of firing these DDK-bound origins prematurely in G1, inevitably leading to re-licensing and re-firing. Eukaryotes control origin firing by DDK and CDK that both increase in activity upon S phase entry to mediate S phase specificity of firing. The requirement for two distinct S phase kinase pathways is considered a safety mechanism against premature firing and bypassing one control pathway likely compromises robustness of this regulation. This means either that origin-bound DDK needs to be kept inactive or that other firing factors need to be strictly suppressed in G1. It was suggested that a protein complex formed by Rif1 and protein phosphatase 1 (PP1) may keep DDK bound to G1 origins in check until the time is right for firing in S phase [117]. This was concluded from the observation that phosphorylation of the DDK substrate Mcm4 increased in G1 phase upon attenuation of PP1 action by inactivating its targeting factor Rif1.

Despite this Rif1-dependent DDK inhibition in G1, there is indication that DDK is active in G1 phase at some early firing origins in budding yeast. Thorough ChIP experiments showed that Sld3^Treslin^ is bound to early origins in G1 phase [65,73]. This suggests that DDK is active at these origins as DDK is required for Sld3^Treslin^ interaction with pre-RCs [13,63,64]. The issue of regulating DDK activity in G1 cells and how origin firing is prevented in G1 to avoid re-replication and genome instability clearly requires further investigation.

Ctf19-mediated early origin firing may have evolved to meet a specific requirement of the pericentromeric genome region. The recruited DDK does not only allow pericentromeric origins to fire early it also recruits the Scc2-Scc4 cohesion loading complex to pericentromeres [38]. In the absence of DDK-dependent Scc2-Scc4 recruitment pericentromeric sister chromatid cohesion was compromised. Thus, Ctf19-mediated DDK recruitment couples early replication with cohesion establishment at centromeres. Why this coupling is necessary is less clear. It is conceivable that the limited time between chromosome duplication and segregation due to short G2 phase in budding yeast necessitates particularly effective control of cohesion establishment at sister centromeres.

### 5.6. Rif1 and Taz1 Facilitate Late Origin Firing in Yeasts

In theory, the late firing of the heterochromatic origins that are not regulated by Swi6/Ctf19 could be a passive process resting on two elements: the competition between origins for limiting firing factors and the low accessibility of heterochromatin for firing factors. These could combine into a delay of firing time. Although this passive process likely contributes to late firing, yeasts employ active regulation to mediate late firing of origins, for example in subtelomeric heterochromatin (Figure 4). The late timing of these origins depends on the proteins Rif1 and Taz1, as genome-wide replication timing analyses of fission yeast mutants showed [112,113]. Budding yeast Rif1 binds PP1 directly through a conserved interaction motif and this is thought to target PP1 to late replicating chromatin [117,118,119]. A model was proposed that binding of fission yeast Rif1-Taz1 complexes to telomeres (Taz1 is homologous to metazoan telomere repeat factors TRF1/2) creates a cloud of PP1 activity around telomeres, delaying the firing of origins in the vicinity, including those in subtelomeres, by counteracting DDK [114]. Supporting this model, genetic interaction studies in budding yeast showed that Rif1 counteracts DDK in delaying origin firing [113,117].

Another class of late firing origins also depends on Taz1 [112]. These origins are located away from telomeres in the chromosome arms and contain telomere repeat DNA for binding Taz1 (Figure 4). About 50% of late origins in chromosome arms belong to this class. Mutant yeast strains showed that the late firing of these origins, unlike subtelomeric origins, requires the shelterin telomere binding complex that interacts with Taz1 [114]. The model developed proposes that Taz1-shelterin interaction brings Taz1-bound chromosome arm origins in close proximity to telomeres, where their firing is delayed by the high local activity PP1 enriched there by Rif1. Consistently, fluorescently labelled Taz1-dependent arm origins are located at the nuclear periphery [114], where telomeres reside due to their anchoring to the inner nuclear membrane [120]. Peripheral localisation of origins is dispensable for late firing as shown using yeast mutants that cannot tether telomeres to the nuclear membrane [114]. Instead, it is probably the proximity of the origins to telomeres mediated by the Taz1-shelterin interaction that delays firing.

Yet another class of late firing origins in chromosome arms depend on Rif1 but not Taz1 [112,113,121]. Rif1 localises in the vicinity of these origins and prevents DDK action, presumably by recruiting PP1 [112,113] (Figure 4).

Together, it appears that yeast Rif1, Taz1 and PP1 constitute a set of versatile firing time regulators that integrate various signals to mediate late firing of different classes of origins.

### 5.7. Metazoa May Utilise Rif1-Dependent and Independent Pathways to Mediate Late Replication

Our picture of the molecular pathways that control origin firing timing in higher eukaryotes is far less complete than in yeasts. Rif1 could be confirmed as an important determinant of late replication. Mouse Rif1 knock-out cells and Rif1 siRNA-treated cultured human cells showed gross changes in genome-wide replication timing profiles [115,122]. The fraction of the genome that changed replication timing was significant: it was greater than the fraction that changes timing during differentiation of embryonic stem cells into neural progenitor cells [115]. A subpopulation of usually late replicating domains shifted to earlier timing and some early domains replicated later. The primary effect of Rif1 inactivation is probably the relieving of firing suppression in late domains, resulting in their earlier replication, whereas the replication delay of early domains is indirect due to increased competition for limiting firing factors. Mouse Rif1 was shown to associate with late replicating domains by genome-wide ChIP and fluorescence microscopy [115]. In a subset of these domains, in chromocenters whose replication dynamics can be particularly well observed microscopically, it appeared that replication only started after Rif1 had left this region. This is consistent with a model where Rif1 inhibits origin firing until its timed departure allows firing.

Two molecular mechanisms have been suggested for how Rif1 mediates late firing in metazoa. (1) Rif1 may delay origin firing by targeting PP1 to late firing origins, similar to yeast. In line with this model, mouse Rif1 associates with PP1 [123]. Moreover, human and *Xenopus* Rif1 and PP1 cooperate in counteracting DDK phosphorylation of Mcm2-7 subunits at replication origins, decreasing their firing efficiency [124]. However, it has not been finally proven that metazoan Rif1 controls firing timing by targeting PP1 to origins to reverse DDK phosphorylation. (2) Rif1 may—alternatively or additionally—control firing time by affecting chromatin domain organisation. 4C-seq analysis revealed that Rif1 deletion in mouse embryonic stem cells and embryonic fibroblasts leads to robust changes in chromatin domain organisation [125]. After Rif1 inactivation, replication domains interacted weaker with neighbouring domains that have the same replication timing in wild-type conditions. Instead, the domains showed a higher number but weak interactions with domains of different replication timing. This change in domain organisation upon Rif1 inactivation was independent of and preceded the change in replication timing, as observed after acute inactivation of Rif1 in synchronised cells. The data was interpreted such that Rif1 is required to define and constrain chromatin domains as physical units of replication timing. In Rif1 knock-out cells these physical units become less defined and as a result the timing of replication changes. This work showed for the first time that the controls of chromatin domain organisation and replication timing are combined in the same molecule, Rif1. Together these observations are consistent with but do not prove, the model that domain structure underlies replication timing.

Also metazoa seem to have more than one molecular pathway to determine replication timing, since upon Rif1 deletion not all late replication domains advance replication time [115]. Two classes of late replicating genomic domains could be defined, one characterised by high levels of Rif1 and association with lamin B (Rif1/LB^+^), the other one characterised by high Rif1 levels but low lamin B (Rif1/LB^−^) [125]. Rif1/LB^-^ domains were the late regions that changed their replication time to early upon Rif1 inactivation, whereas Rif1/LB^+^ domains did not. This means that Rif1 is required for late replication of one class of domains, Rif1/LB^−^. In Rif1/LB^+^ domains, Rif1 is either not required to regulate timing or it acts redundantly with another timing control mechanism.

A third class of late domains may be controlled by polymerase theta [126]. In polymerase theta depleted cells, earlier replication of a subset of late domains was observed. This was accompanied by higher levels of Mcm2-7 helicase on chromatin, suggesting that increased licensing may underlie earlier replication. It is conceivable that excess pre-RCs advance replication timing by increasing the local concentration of firing factors or by inducing a more open chromatin structure.

In conclusion, a model is emerging that, like yeasts, higher eukaryotes utilise distinct mechanisms to control replication timing of different genome regions.

### 5.8. The TMT Complex May Help Control Firing Timing in Metazoa

The TMT complex, Treslin/TICRR^Sld3^-MTBP^Sld7^-TopBP1^Dpb11^, is a regulation platform for origin firing in metazoa. Its counterpart in yeast, the SDS complex, Sld3^Treslin^-Sld7^MTBP^-Dpb11^TopBP1^ together with Sld2^RecQL4^, mediates cell cycle and DNA damage checkpoint regulations and these factors are subject to subtle control of levels and activity to ensure complete and accurate genome replication [12,13,127,128]. In metazoa, TMT mediates conserved cell cycle and checkpoint regulations [83,84] and TopBP1^Dpb11^ and Treslin/TICRR^Sld3^ have also been implicated in other regulations, for example the coupling of genome replication to sufficiently high nutrient levels [129]. The TMT proteins are well-equipped to integrate signals from many regulatory pathways. Apart from the conserved core domains Treslin/TICRR^Sld3^ contains metazoa-specific N- and C-terminal domains that do not perform essential core origin firing functions but are required for normal DNA replication in human cells [82,94,95] (Ferreira and Boos, unpublished). Moreover, MTBP whose yeast orthologue Sld7^MTBP^ does not seem to be a very important signal integrator has apparently evolved into an origin firing regulator in metazoa. Apart from the Sld7-homologous functions, it contains a metazoa-specific domain that is important for complete genome duplication in human cells [89]. The molecular functions of this domain are only partially understood.

The metazoa-specific C-terminus of Treslin/TICRR^Sld3^ has been implicated in controlling replication timing [95]. A short motif in the large C-terminus mediates the binding of Treslin/TICRR^Sld3^ to the BRD2 and BRD4 proteins that associate with acetylated chromatin via bromodomains. This interaction recruits Treslin/TICRR^Sld3^ to chromatin, presumably at acetylated genome regions. Treslin/TICRR^Sld3^ mutants that cannot bind to BRD2/4 replicated at relatively normal overall speed but showed aberrant execution of the replication program. EdU labelling and fluorescence microscopy of human cells revealed that early and mid S phase replication foci patterns merge in cell lines expressing BRD2/4 binding-deficient Treslin/TICRR^Sld3^. These aberrant patterns suggest changed replication timing. Genome-wide timing analysis in these mutants will be required to confirm and specify the timing changes. A feasible model of how Treslin/TICRR^Sld3^ mediates correct replication timing is that its appropriately timed acetylation-dependent recruitment to specific chromatin sites induces replication locally. How the appropriate timing of Treslin/TICRR^Sld3^ recruitment is regulated in the cell is a key question. Because acetylation is often associated with open euchromatin that replicates early Treslin/TICRR^Sld3^ recruitment could be important for early replication and, as a consequence of competition for limiting firing factors, help to delay replication of late domains, which could explain the observed merging of early and late patterns in cells lacking this regulation.

## 6. Firing Coherence Mechanisms

As already mentioned, it is important for cells to fire enough origins in a short time window once a replication domain is set to replicate. How such coherent origin firing is achieved has been little investigated. It must involve a local, switch-like transition from an origin firing-incompetent to a competent state to ensure that, before the switch is turned, not a single origin fires and, after turning, many origins fire nearly synchronously. Thus, the rate-limiting step(s) of the origin firing reaction must be switched on both sharply and locally. This switch could involve any of the molecular mechanisms used to control the localisation or activity of firing factors discussed above. Intuitively, linking the origins of a domain, for example, by physical clustering, like suggested for forkhead factors [111], could be the basis to create coherent firing. Such physical linkage between origins could help control the firing threshold of origins in a cluster. It was proposed that it would suffice to have a higher threshold for the firing of the very first origin of a replication domain than for the others to create nearly synchronous firing in a replication domain [130].

Mathematical modelling suggested that, in yeast, the multisite phosphorylation of Sld2^RecQL4^ and Sld3^Treslin^ could induce their switch-like binding to Dpb11^TopBP1^ in a manner, mediating coherent origin firing [131]. This is reminiscent of a famous example of biological switches, the CDK inhibitor Sic1. In response to rising CDK levels at the G1-S phase transition, six CDK phosphorylation sites of Sic1 act cooperatively to induce the switch-like degradation of Sic1 by the ubiquitin-proteasome pathway [132]. In analogy, the origins of a replication domain could be switched on when local CDK activity passes a threshold between an intermediate level that does not promote firing but inhibits licensing and a high level that allows firing. Although the CDK phosphorylation of only two sites in Treslin/TICRR^Sld3^ is sufficient for relatively high affinity binding to TopBP1^Dpb11^, Treslin/TICRR^Sld3^ contains dozens more CDK consensus sites that could regulate this binding event.

## 7. Conclusions

Eukaryotes have evolved distinct molecular pathways to control firing timing of distinct sets of origins. This indicates that appropriate replication timing serves to adapt genome replication to the specific requirements of the different genomic regions. These active timing control pathways combine with the competition of origins for limiting firing factors to establish the temporal program of genome replication. During competition between origins the much-discussed intrinsic firing propensity of individual origins likely plays an important role in determining which origins win the competition and fire first.

Pre-IC formation is the step at which firing timing regulation occurs. In yeasts, the control of DDK activity is a common theme among firing control pathways. In higher eukaryotes, the TMT regulation platform is a good candidate to mediate firing time control. Rif1 is conserved from yeast to man as a mediator of late firing. Much research is required, particularly in metazoa, to uncover and understand how origin firing time is determined.

Once the molecular mechanisms are known it will be possible to directly address how important replication timing control is by manipulating these pathways using genetics and chemical inhibitors. As mentioned, replication timing changes with cell differentiation and tumorigenesis and has been linked to genome instability but the nature of these links—e.g., if timing changes are cause or consequence and what the cellular mechanisms are—is mostly unknown. Thus, investigating the relevance of replication timing control is important for our understanding of how cells work and also for human health.

## Figures and Tables

**Figure 1 genes-10-00199-f001:**
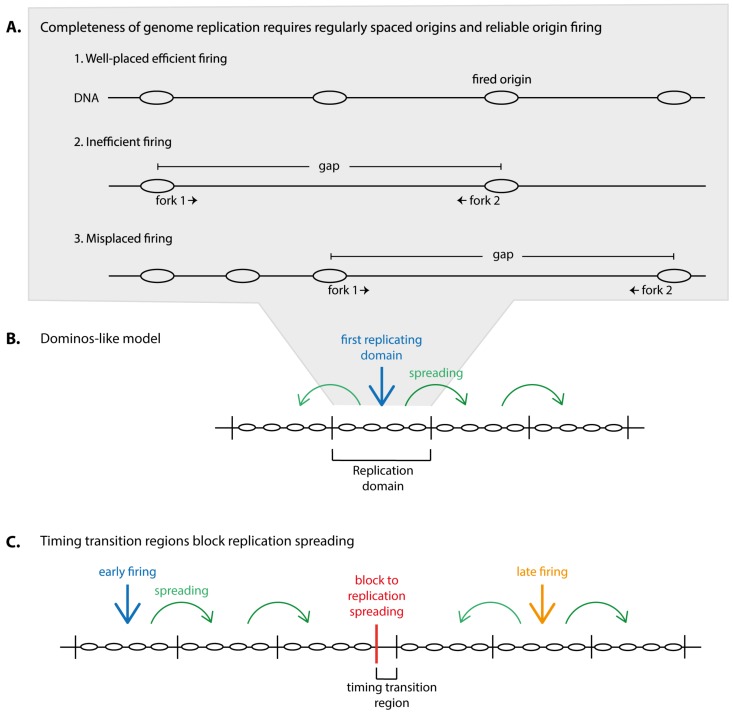
Proper control of origin firing is required for complete genome duplication. (**A**) Well-placed origins and efficient firing are required for complete DNA replication (1.) Replication problems can arise if not enough origins fire (2.) or if origins are misplaced (3.) Such misregulations can result in gaps in the replicating genome that are too big to be replicated by two forks during an S-phase. (**B**,**C**) Schematic representation of the dominos-like model in which replication spreads from the first replicating domain to its neighbours. This replication spreading is blocked at timing transition regions (**C**). Timing of the later domain at timing transition regions (TTRs) requires a new timing signal.

**Figure 2 genes-10-00199-f002:**
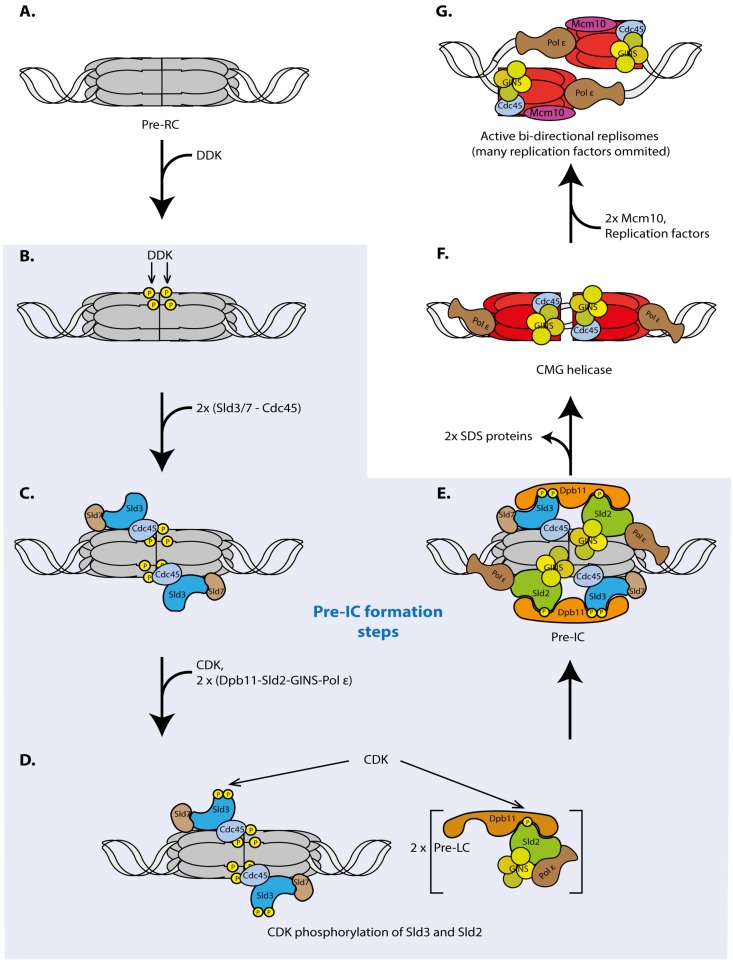
Molecular mechanisms of origin firing. Schematic representation of origin firing—a pre-RC (**A**) being converted into two active bi-directional replisomes (**H**)—and the steps required (**B**–**G**), as detailed in the main text. The blue box indicates pre-IC formation (**B**–**F**). Some details shown are hypothetical. For example, we speculate that two pre-IC complexes are required to originate two bi-directional forks but other models are also possible.

**Figure 3 genes-10-00199-f003:**
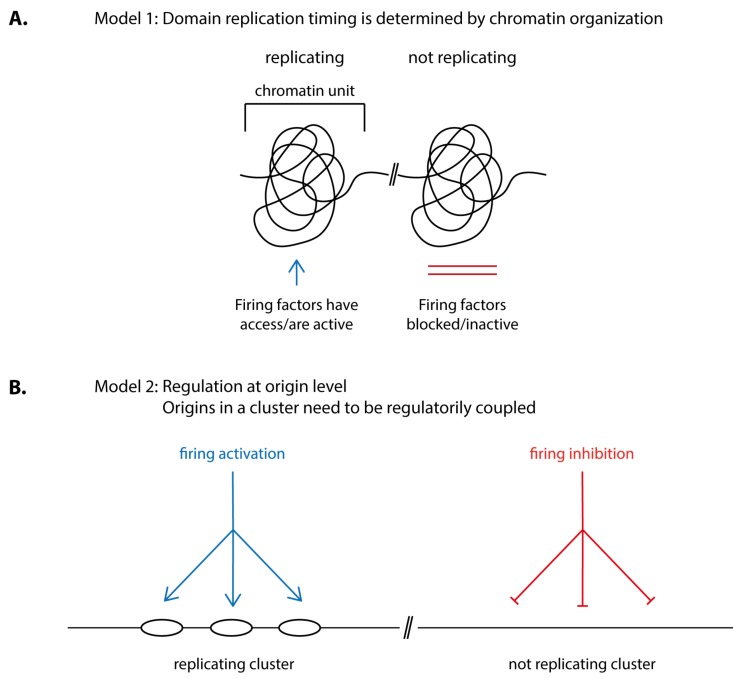
Two models of origin firing control to determine replication timing. (**A**) In model 1, origin firing timing is determined by chromatin organization. Different architectural chromatin units, depending on their structure, allow access/activate or block/inactivate firing factors, defining the timing of origin firing. Origins within a unit fire nearly synchronously when firing is allowed. (**B**) In model 2, origin firing timing is regulated at the origin level. This postulates that origins in a cluster fire synchronously because they are regulatorily coupled. Activating and inhibiting regulations define whether a cluster of origins fires or not at a given time.

**Figure 4 genes-10-00199-f004:**
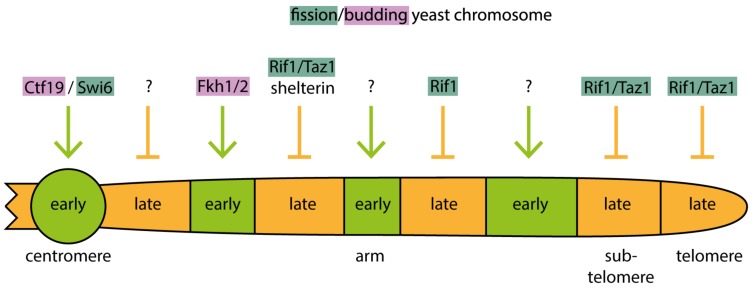
Origin firing pathways that control replication timing of yeast chromosomes. Work in fission and budding yeast has revealed several molecular pathways that determine whether a chromosomal region replicates early (green) or late (orange). Ctf19/Swi6 promotes early replication of the centromere, while Fkh1/2 are involved in the early replication of some regions of the chromosome arm. Rif1 and/or Taz1 and/or shelterin dependent regulations ensure the late replication timing of the telomeric, sub-telomeric and some chromosomal arm regions. Other uncharacterised mechanisms for early or late replication timing regulations exist and are indicated with a question mark.

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
