# Peer review of "Origin Firing Regulations to Control Genome Replication Timing"

_genes, 2019, doi:10.3390/genes10030199_

Round 1

Reviewer 1 Report

Comments on genes-447212 (Dominik Boos and Pedro Ferreira, Origin Firing Regulations to Control Genome Replication Timing).

In this review article, authors review how the firing of multiple replication origins is regulated in eukaryotes. They focus mainly on the temporal regulation of origin firing. Generally, this review is well-written and therefore worth to publish in the genes. However, I feel there are several issues to be clarified before publication. If they are solved, I recommend the publication of this review article. Specific points are follows.

Major points

1. In this review, it looks like authors construct their story based on the idea in which all replication origins fire. However, it is known that excess origins are licensed and some of them do not fire under the normal context (‘dormant’ origins). Such dormant origins fire to ensure the un-replicated regions are not left, when replication fork did not reach to them. This is a well-known fact and the one of the ways to ensure faithful DNA replication. If authors include this point in some organized way in their review, it would be very nice.

2. Authors’ recognition of the detailed mechanisms of the firing reaction seems to be slightly different from the widely-accepted model. For example, authors describe ‘The first step of origin firing is phosphorylation of pre-RCs by DDK, which leads to the binding of the Sld3-Sld7 protein complex’ (lines 140-141). This would be true for the in vitro reconstituted system, however, in vivo, Sld3-(Sld7)-Cdc45 are loaded to Mcm2-7 in mutually dependent manner, which is originally shown by Kamimura et al (EMBO J, 2001) and further confirmed by Miyazawa-Onami et al (EMBO R, 2017). To be honest, I am surprised that authors did not cite the Kamimura et al paper and the Sld7 paper (Tanaka et al EMBO J, 2011).

  They also describe ‘As a result, the so-called SDS complex (Sld3/7-Dpb11-Sld2) is formed’ (line 145). However, as far as I know, no one have detected such complex alone and the word ‘SDS complex’ are not used in the field. Related to these, Figure 2 need to be fixed, although the Figure does not seem to have a lot of problems. I think the complex described between C and D and the SDS complex between F and G have never isolated and described. It has also shown that all the firing factors those are recruited to replication origins in CDK-dependent process associate to replication origins in mutually dependent manner (Miyazawa-Onami et al, EMBO Rep, 2017). This suggests the existence of the SDS complex alone is unlikely.

 Based on the argument of the SDS complex, authors describe TMT complex. Is the existence of TMT complex alone (not the complex including the T-M-T) proved? If it is, please describe.

 Actually, in the section 3, authors cited previous studies inaccurate or inappropriate way. For example, in the in vitro reconstitution by Yeeles et al (Nature, 2015), CDK, DDK, Sld2, Sld3, Sld7, Dpb11, Cdc45, the GINS complex,

Pol ε and Mcm10 are shown to sufficient to activate replicative helicase and further addition of Pol α, Ctf4, RPA, and Topo II generated newly synthesized both leading and lagging strand DNA (although the effect of Ctf4 is minor). Therefore, authors’ description in lines 129-130 seems incorrect for me.  Moreover, authors describe that the CMG complex assemble AFTER the pre-IC formation (lines 148-149). However, as shown by van Deursen et al (EMBO J 2012) and Watase et al (Curr Biol 2012), stable CMG complex is observed in the Mcm10-depleted cells, and the Mcm2-7 double hexamer is already separated at this point (Miyazawa-Onami et al EMBO Rep, 2017; Douglas et al Nature 2018). Moreover, the first description of the CMG is Moyer et al (PNAS 2006) and Gambus et al (Nat Cell Biol 2006), not Ilves et al (Mol Cell 2010) nor Labib et al (Science 2000).

3. I do not think the control of origin firing by DDK and CDK are ‘redundant’ (lines 347-348). CDK and DDK regulate origin firing by controlling the distinct step. 

Minor points

1. Some of their expressions are vague and hard to get the point for me. For example:

 Fundamental principles of eukaryotic origin are known (line15); 

 many regulation pathways culminate… (line 21);

 in order to minimise the number of large inter-origin gaps (line 36-37);

 Because of the higher origin numbers … replication (lines 45-49);

 If the relatively rare firing events … ensue (lines 53-54);

 The timing with which … to be determined (lines 114-115)

 All temporal and spatial regulations … must ultimately culminate (lines 123-124);

 Pre-IC-level regulation is not … invokes pre-IC factors (lines 174-176)

 causal poof (line 244).

2. Authors describe the relationship between chromatin regulators and origin firing through origin licensing in metazoa. They missed to cite some previous studies on Hbo1, SNF2, GRWD1, and PR-Set7. Lines 269-269 and 444-447 seem redundant. There might be another redundancy.

3. Line 358. Reference should be Kamimura et al., and Line 359. Reference #3 should be added.

 There might be another improper citation, author should carefully go through the whole manuscript again.

4. ‘Köhler et al accepted’ (lines 166, 185, 189) and ‘Köhler and Boos, accepted’ (line 464), please include the journal name or more details, if possible.

Author Response

Response to reviewer 1

We greatly appreciate this thorough review. It has contributed significantly to represent the current knowledge in a more balanced way.

Reviewer 1 feels that our review article about origin firing regulations with focus on temporal regulation is worth publishing if some points raised are addressed. The reviewer is concerned about an underappreciation of the fact that an excess of origins are licensed but only few fire. This is a point also raised by reviewer 2 and we have addressed this. Reviewer 1 also raises some points regarding our description of replication initiation processes. The underlying concern seems to be that our view on initiation based on recent in vitro reconstitution is too narrow and underappreciates the in vivo situation. We thank the reviewer for the helpful remark and have amended this in the revised manuscript. The reviewer also suggests some changes of citations. We also thank the reviewer for these good suggestions. Reviewer 1 also asks for integrating the involvement of several proteins with relation to chromatin structure in replication initiation, mainly licensing, in vivo. Although this is listed as a minor point to be addressed reviewer 2 raises a similar questions. We have, thus, addressed it in the revised manuscript.

Point-by-point response

Major points

1. In this review, it looks like authors construct their story based on the idea in which all replication origins fire. However, it is known that excess origins are licensed and some of them do not fire under the normal context (‘dormant’ origins). Such dormant origins fire to ensure the un-replicated regions are not left, when replication fork did not reach to them. This is a well-known fact and the one of the ways to ensure faithful DNA replication. If authors include this point in some organised way in their review, it would be very nice.

We thank the reviewer for this suggestion that allows a broader perspective of genome replication. We added from line 54: “One important mechanism utilized by eukaryotes to reduce large inter-origin distances that is not discussed in detail in this article is the use of dormant origins [2]. Eukaryotic cells generate many more potential origins by origin licensing (discussed below) than actually fire in normal S phases [3-8]. These are dormant origins that fire only if they have enough time, as shown in yeast and vertebrates [9-11]. Unfired dormant origins will normally get inactivated if their DNA is replicated by an incoming fork generated by a neighbouring origin. If this does not occur, for example when no origin in the vicinity fires or when DNA damage stalls incoming forks, dormant origin firing helps complete duplication of the DNA in their genome region.

In addition to dormant origins that help reduce size of many inter-origin gaps eukaryotes have also evolved a complex genome replication program to ensure complete genome replication, which is discussed in detail in this review.

2. Authors’ recognition of the detailed mechanisms of the firing reaction seems to be slightly different from the widely-accepted model. For example, authors describe ‘The first step of origin firing is phosphorylation of pre-RCs by DDK, which leads to the binding of the Sld3-Sld7 protein complex’ (lines 140-141). This would be true for the in vitro reconstituted system, however, in vivo, Sld3-(Sld7)-Cdc45 are loaded to Mcm2-7 in mutually dependent manner, which is originally shown by Kamimura et al (EMBO J, 2001) and further confirmed by Miyazawa-Onami et al (EMBO R, 2017). 

To appreciate this point we have added from line 172 including the relevant citations:

“In vivo, the situation might be slightly more complicated as Sld3 and Cdc45 recruitment to origins depend on each other {Kamimura, 2001 #8248}{Miyazawa-Onami, 2017 #12525}. 

To be honest, I am surprised that authors did not cite the Kamimura et al paper and the Sld7 paper (Tanaka et al EMBO J, 2011).

The reviewer is completely right. We added Kamimura et al in line 173, and Tanaka et al in line 218. We apologise particularly to Tanaka et al. The citation must have accidentally fallen out.

They also describe ‘As a result, the so-called SDS complex (Sld3/7-Dpb11-Sld2) is formed’ (line 145). However, as far as I know, no one have detected such complex alone and the word ‘SDS complex’ are not used in the field. Related to these, Figure 2 need to be fixed, although the Figure does not seem to have a lot of problems. I think the complex described between C and D and the SDS complex between F and G have never isolated and described. It has also shown that all the firing factors those are recruited to replication origins in CDK-dependent process associate to replication origins in mutually dependent manner (Miyazawa-Onami et al, EMBO Rep, 2017). This suggests the existence of the SDS complex alone is unlikely.

To account for this remark we changed Figure 2: The complex between C and D was taken out. We also included in the manuscript from line 177:

As a result, the so-called SDS complex (Sld3/7-Dpb11-Sld2) may form. Although phospho-Sld2 and phospho-Sld3 interactions with Dpb11 events are essential for origin firing [69, 70] it is possible that, in vivo, they do not bind simultaneously or bind transiently because a stable SDS complex in cells has not been reported. Sld2 associates with and recruits GINS and DNA polymerase epsilon to pre-ICs [71], whilst Sld3 recruits Cdc45 [63]. We here call the SDS complex, Cdc45, GINS and polymerase epsilon bound to pre-RCs the pre-initiation complex (pre-IC) [66, 72], although the exact structure of the pre-IC remains unclear. Pre-IC can be stably assembled in vitro [41, 64] and can be detected by ChIP in vivo [66] (Figure 2E). Cdc45 and GINS assemble stably [73] with the Mcm2-7 helicase to form the CMG complex (Cdc45-Mcm2-7-GINS) [56, 74], the active replicative helicase [57, 75] (Figure 2F).

Based on the argument of the SDS complex, authors describe TMT complex. Is the existence of TMT complex alone (not the complex including the T-M-T) proved? If it is, please describe.

The TMT complex is stable enough to be detected in vivo. We included this from line 222: “A stable protein complex containing Treslin/TICRRSld3, MTBPSld7 and TopBP1Dpb11 could be detected in cell lysates [88]. We will call it TMT complex (Treslin-MTBP-TopBP1) from here.”

Actually, in the section 3, authors cited previous studies inaccurate or inappropriate way. For example, in the in vitro reconstitution by Yeeles et al (Nature, 2015), CDK, DDK, Sld2, Sld3, Sld7, Dpb11, Cdc45, the GINS complex, Pol ε and Mcm10 are shown to sufficient to activate replicative helicase and further addition of Pol α, Ctf4, RPA, and Topo II generated newly synthesized both leading and lagging strand DNA (although the effect of Ctf4 is minor). Therefore, authors’ description in lines 129-130 seems incorrect for me.

What reviewer 1 writes does not seem to contradict our statement. Yeeles et al 2015 show in Fig4 that in the absence of any of the factors CDK DDK, Sld2, Sld3, Sld7, Dpb11, Cdc45, the GINS complex, DNA polymerases epsilon and alpha/primase, and Mcm10, no in vitro DNA synthesis can be observed. Combining all of these factors supports in vitro replication. Thus, our statement that these factors are sufficient for leading and lagging strand synthesis is correct. RPA, Ctf4 and TopoII are important but not essential in vitro. We therefore call the essential core initiation factors sufficient to generate bi-directional leading/lagging strand synthesis.

Moreover, authors describe that the CMG complex assemble AFTER the pre-IC formation (lines 148-149). However, as shown by van Deursen et al (EMBO J 2012) and Watase et al (Curr Biol 2012), stable CMG complex is observed in the Mcm10-depleted cells, and the Mcm2-7 double hexamer is already separated at this point (Miyazawa-Onami et al EMBO Rep, 2017; Douglas et al Nature 2018). 

We agree that CMG forms Mcm10-independently and state this in line 191: “After CMG formation active replisomes assemble in an Mcm10 dependent manner [41, 76-78]…”

The reviewer is correct that, formally, the time point of CMG formation, as defined by Kanemaki et al 2006 as salt-resistant Cdc45 association, may not be quite clarified. We imagine that this occurs after or during dissociation of the SDS proteins from origins (Figure 2). Because - according to our definition of pre-IC - SDS is part of pre-IC we write CMG forms AFTER pre-IC. We acknowledge that this is hypothetical and that CMGs could already form in pre-ICs. We therefore deleted “Subsequently” (indicating AFTER) in line 184.

Moreover, the first description of the CMG is Moyer et al (PNAS 2006) and Gambus et al (Nat Cell Biol 2006), not Ilves et al (Mol Cell 2010) nor Labib et al (Science 2000).

Our citation is correct but we now also added Moyer et al and Gambus et al in line 185.

3. I do not think the control of origin firing by DDK and CDK are ‘redundant’ (lines 347-348). CDK and DDK regulate origin firing by controlling the distinct step. 

We deleted ‘redundant’ and changed our text from line 422: “Eukaryotes control origin firing by DDK and CDK that both increase in activity upon S phase entry to mediate S phase specificity of firing. The requirement for two distinct S phase kinase pathways is considered a safety mechanism against premature firing, and bypassing one control pathway likely compromises robustness of this regulation.”

Minor points:

1. Some of their expressions are vague and hard to get the point for me. For example:

 Fundamental principles of eukaryotic origin are known (line15); 

The word “firing” had accidentally been left out and has been added now. This should clarify the issue

 - many regulation pathways culminate… (line 21);

“culminate in” was deleted.

- in order to minimise the number of large inter-origin gaps (line 36-37);

The sentence was changed into (line 36): “In order to keep the number of large inter-origin gaps minimal to guarantee completeness of genome duplication two requirements must be met…”

- Because of the higher origin numbers … replication (lines 45-49);

The sentence was changed into (line 51): “Because larger genomes have more origins the total number of very large inter-origin distances that can potentially not be completely replicated rises with genome size.”

- the relatively rare firing events … ensue (lines 53-54);

We are unsure what is unclear in this sentence. Firing is “rare” because, as stated in the previous sentence, not all origins fire at the same time due to limiting firing factors. We would appreciate suggestions.

- The timing with which … to be determined (lines 114-115)

In line 135, “determined” was exchanged for “controlled”. Does this clarify the issue?

- All temporal and spatial regulations … must ultimately culminate (lines 123-124);

The sentence now reads (from line 150): “All temporal and spatial regulations of replication origin firing must ultimately control the activity of essential origin firing factors.”

- Pre-IC-level regulation is not … invokes pre-IC factors (lines 174-176)

The sentence now reads (from line 231): “Regulation at the level of pre-IC formation is not a cell cycle-specific feature but rather a principle of origin firing control, as exemplified by the fact that firing regulation upon DNA damage also invokes pre-IC factors.”

- causal poof (line 244)

The sentence now reads (from line 314): “However, direct poof that the structuring of chromatin into folding units underlies the determination of replication timing has not been provided.”

2. Authors describe the relationship between chromatin regulators and origin firing through origin licensing in metazoa. They missed to cite some previous studies on Hbo1, SNF2, GRWD1, and PR-Set7. Lines 269-269 and 444-447 seem redundant. There might be another redundancy.

We thank the reviewer for this remark. From line 163 is now included: “The core licensing machinery sufficient for pre-RC formation on naked origin DNA in vitro constitutes ORC (origin recognition complex), Mcm2-7, Cdt1 and Cdc6 [43, 44]. Other factors influence licensing of chromatin in living cells, such as the chromatin factors Hbo1 [45, 46], SNF2 [47], GRWD1 [48], and PR-Set7 [49-53], or Mcm-BP [54, 55].”

3. Line 358. Reference should be Kamimura et al., and Line 359. Reference #3 should be added.

There might be another improper citation, author should carefully go through the whole manuscript again.

Done

4. ‘Köhler et al accepted’ (lines 166, 185, 189) and ‘Köhler and Boos, accepted’ (line 464), please include the journal name or more details, if possible.

The paper is now out and properly cited in the revised manuscript.

Reviewer 2 Report

Boos and Ferreira reviewed the how genome replication is controlled by regulations of origin firing in higher eukaryotes, such as yeast, human and Xenopus.

Comments:

MCM proteins play a key regulatory role in origin firing.

-MCM paradox: MCM is more abundant compare to origins of replication.

-MCM-BP (MCM binding protein) is an important factor contribute to Pre-RC formation. It dissociates MCM complex from chromatin.

It will be helpful if there is a table listing all the molecules involved in the replication complexes formation and their functions

How do replication forks deal with DNA damages? What happens when a replication fork collapsed? Do late origins play a role in forming new origins to restart?

It might be good to talk a little about the players involved in chromatin remodeling/dynamic in DNA replication.

Author Response

Response to reviewer 2

We greatly appreciate this review.

Point-by-point response

1) MCM proteins play a key regulatory role in origin firing.

A) MCM paradox: MCM is more abundant compare to origins of replication.

We appreciate this point in the revised manuscript by adding from line 54: 

One important mechanism utilized by eukaryotes to reduce large inter-origin distances that is not discussed in detail in this article is the use of dormant origins [2]. Eukaryotic cells generate many more potential origins by origin licensing (discussed below) than actually fire in normal S phases [3-8]. These are dormant origins that fire only if they have enough time, as shown in yeast and vertebrates [9-11]. Unfired dormant origins will normally get inactivated if their DNA is replicated by an incoming fork generated by a neighbouring origin. If this does not occur, for example when no origin in the vicinity fires or when DNA damage stalls incoming forks, dormant origin firing helps complete duplication of the DNA in their genome region.

In addition to dormant origins that help reduce size of many inter-origin gaps eukaryotes have also evolved a complex genome replication program to ensure complete genome replication, which is discussed in detail in this review.

B) MCM-BP (MCM binding protein) is an important factor contribute to Pre-RC formation. It dissociates MCM complex from chromatin.

We address this point in the revised manuscript from line 165: Other factors influence licensing of chromatin in living cells, such as the chromatin factors Hbo1 [45, 46], SNF2 [47], GRWD1 [48], and PR-Set7 [49-53], or Mcm-BP [54, 55]. 

2) It will be helpful if there is a table listing all the molecules involved in the replication complexes formation and their functions

The factors are named in Figure 2. We feel that, for many of the factors, too little is known about their functions to describe them in a few key words of a table. Figure 2 and the main text provide a more balanced representation.

3) How do replication forks deal with DNA damages? What happens when a replication fork collapsed? Do late origins play a role in forming new origins to restart?

To account for this point we have included in the revised manuscript the function of dormant origins using the text in 1A. We do not want to discuss the interference of DNA lesions with replisomes in much detail because we focus in our review on replication timing.

4) It might be good to talk a little about the players involved in chromatin remodeling/dynamic in DNA replication.

We appreciate this comment as detailed in point 1B.

Reviewer 3 Report

It was a great pleasure to read the review "Origin firing regulations to control genome replication timing". The review handled well both the detail and the broad scope of the title and it will be a useful and timely summary for the field. I have some minor comments, which should be considered before publication.

in line 72/73 is not good English

Line 146/147 P.5. I am not sure that the definition of pre-initiation is quite correct. seeZou L and Stillman B (1998) Formation of a preinitiation complex by S‐phase cyclin CDK‐dependent loading of Cdc45p onto chromatin.Science, 280, 593596.

Line 62 says " metazoan Sld2..... does not seem to depend on CDK". but C.elegans Sld2 is an essential CDK target. J Cell Biol. 2014 CDK phosphorylation of SLD-2 is required for replication initiation and germline development in C. elegans.Gaggioli V, Zeiser E, Rivers D, Bradshaw CR, Ahringer J, Zegerman P.

Line 180 should say inhibits rather than inactivates, as the mechanism for Rad53 regulation of DDK is not clear.

Line 210. Not sure that origins can be described as 'sensitive' to limiting factors. Maybe accessible? greater affinity?

Line 331. Should Dfb1 be Dfp1?

7. line 443/444 polymerase theta hasn't formatted properly

Author Response

Response to reviewer 3

Reviewer 3 feels that our manuscript provides a timely and balanced representation of the field, and suggests some minor changes. We thank reviewer 3 for her/his useful comments.

Point-by-point response

1) in line 72/73 is not good English

We changed these sentences to (from line 89) to: “This may minimise the frequency with which domains that replicate in isolation occur, that is domains replicating at vastly different times from their neighbours. Such isolated domains inevitably generate two forks, the two outmost forks, that will find opposing forks for termination only with a significant time delay.

2) Line 146/147 P.5. I am not sure that the definition of pre-initiation is quite correct. see Zou L and Stillman B (1998) Formation of a preinitiation complex by S‐phase cyclin CDK‐dependent loading of Cdc45p onto chromatin.Science, 280, 593–596.

Zhou and Stillman defined pre-ICs as an intermediate of initiation that depends on CDK activity. They used Cdc45 association as a marker for pre-ICs. We think that our definition (Fgure 2) is a specification of this initial definition rather than an alternative. This specification has become possibly by identification and characterisation of Sld2, Sdl3, Dbp11, GINS, and the role of CDK in origin firing. These were not known or largely uncharacterised at the time of the Stillman definition. Our definition fits well with that of other labs in the field (Miyazawa-Onami et al EMBO Rep, 2017)

3) Line 62 says " metazoan Sld2..... does not seem to depend on CDK". but C.elegans Sld2 is an essential CDK target. J Cell Biol. 2014 CDK phosphorylation of SLD-2 is required for replication initiation and germline development in C. elegans. Gaggioli V, Zeiser E, Rivers D, Bradshaw CR, Ahringer J, Zegerman P.

The reviewer is right. Thanks a lot for pointing out this mistake. We changed “metazoan” to “vertebrate”

4) Line 180 should say inhibits rather than inactivates, as the mechanism for Rad53 regulation of DDK is not clear.

done (line 238)

5) Line 210. Not sure that origins can be described as 'sensitive' to limiting factors. Maybe accessible? greater affinity?

We have changed this from line 276 to: “For example, origins of early domains could respond more sensitively than late origins to limiting origin firing factors, for example due to higher accessibility or higher affinity, and therefore fire earlier.”

6) Line 331. Should Dfb1 be Dfp1?

Correct and changed (line 405).

7) line 443/444 polymerase theta hasn't formatted properly

changed (line 522)

Reviewer 4 Report

This is a really well structured, thoughtful and interesting review. 

I don't have any major suggestion for the manuscript. 

My only comments are:

 - do the authors believe worthy to include the observation that Mcm2-7 SUMOylation in budding yeast appears to recruit Rif1 and Glc7, thus regulating the replication timing? Is SUMOylation a potential level of control of origin firing? Nevertheless, I understand that the field is not developed enough to speculate about this possibility.

- theta doesn't appear (lines 443-444). Can the authors speculate about how DNA pol theta controls late origins firing?

Author Response

Response to reviewer 4

Reviewer 4 feels that our review is well structured, thoughtful and interesting.

Point-by-point response

Minor points:

1) do the authors believe worthy to include the observation that Mcm2-7 SUMOylation in budding yeast appears to recruit Rif1 and Glc7, thus regulating the replication timing? Is SUMOylation a potential level of control of origin firing? Nevertheless, I understand that the field is not developed enough to speculate about this possibility.

We also feel that this point not been developed enough. We than the reviewer for pointing this out.

2) - theta doesn't appear (lines 443-444). Can the authors speculate about how DNA pol theta controls late origins firing?

Formatting changed in line 522.

We speculate about the mechanism from line 522: “A third class of late domains may be controlled by polymerase theta [126]. In polymerase theta depleted cells, earlier replication of a subset of late domains was observed. This was accompanied by higher levels of Mcm2-7 helicase on chromatin, suggesting that increased licensing may underlie earlier replication. It is conceivable that excess pre-RCs advance replication timing by increasing the local concentration of firing factors or by inducing a more open chromatin structure.”

Because this is already really speculative we did not feel very comfortable to speculate even more. But we would be open for suggestions by the reviewer.

Round 2
